# Modeling of Charge Injection, Recombination, and Diffusion in Complete Perovskite Solar Cells on Short Time Scales

**DOI:** 10.3390/ma16227110

**Published:** 2023-11-10

**Authors:** Krzysztof Szulc, Katarzyna Pydzińska-Białek, Marcin Ziółek

**Affiliations:** 1Institute of Spintronics and Quantum Information, Faculty of Physics, Adam Mickiewicz University in Poznan, Uniwersytetu Poznańskiego 2, 61-614 Poznan, Poland; krzysztof.szulc@amu.edu.pl; 2Institute of Physics, Faculty of Physics, Adam Mickiewicz University in Poznan, Uniwersytetu Poznańskiego 2, 61-614 Poznan, Poland; katarzyna.pydzinska@amu.edu.pl

**Keywords:** perovskite solar cells, charge carrier transfer, ultrafast spectroscopy

## Abstract

A model of charge population decay upon ultrafast optical pulse excitation in complete, working perovskite solar cells is proposed. The equation, including charge injections (extractions) from perovskite to contact materials, charge diffusion, and charge recombination via first-, second-, and third-order processes, is solved using numerical simulations. Results of simulations are positively verified by broadband transient absorption results of mixed halide, triple-cation perovskite (FA_0.76_MA_0.19_Cs_0.05_Pb(I_0.81_Br_0.19_)_3_). The combined analytical and experimental findings reveal the best approaches for the proper determination of the crucial parameters that govern charge transfer dynamics in perovskite solar cells on picosecond and single nanosecond time scales. Measurements from both electron and hole transporting layer sides under different applied bias potentials (zero and close to open circuit potential) and different pump fluence (especially below 5 μJ/cm^2^), followed by fitting of parameters using numerical modeling, are proposed as the optimal methodology for describing the processes taking place in efficient devices.

## 1. Introduction

Perovskite solar cells (PSCs) have advanced rapidly, with power conversion efficiencies (PCEs) now exceeding 25% [1]. As in most photovoltaic systems, to obtain a high photocurrent of the device, a broad absorption of the active material and efficient photoinduced charge separation to the contact materials (electrons from one side and holes from the other side, Figure 1A) are required. The charge separation in PSCs starts in the sub-ps time range. Therefore, time-resolved laser spectroscopy has to be employed for the investigation of ultrafast charge transfer dynamics [2,3,4,5,6,7,8]. The study of charge transfer in perovskite materials and PSCs is an important topic but also a challenging task [9,10,11]. In principle, the decay of the photoexcited charges in the isolated perovskite material reveals the dynamics of the internal recombination. In complete devices, it is additionally affected by both electron and hole transfer from perovskite to contact materials: electron-transporting (ETM) and hole-transporting material (HTM), respectively, the presence of which accelerates the observed charge population decay [12,13]. The quantum yields of charge transfers to ETM and HTM determine the photocurrent and efficiency of the device and depend on the competition between the charge transfer (charge injection from perovskite to ETM/HTM, often also called charge extraction) rate constants (*k*_IET_ and *k*_IHT_, Figure 1A) and the recombination rate constants. However, under laser pulse excitation, not only the first-order recombination (trap-assisted, with rate constant *k*_1_) takes place, but also second- (radiative, *k*_2_) or even third-order (Auger, *k*_3_) processes are frequently observed, and thus the apparent decay kinetics depend on the pump pulse intensity even for relatively low-intense pulses [14]. Moreover, in perovskite devices, the thickness of the active material is in the order from tens to several hundreds of nanometers, and the initial distribution of the photoexcited charges (Figure 1B) changes due to the charge diffusion process that occurs on the time scales from hundreds of picoseconds to single nanoseconds [14,15,16]. Therefore, the following equation for the charge population *n* has to be solved, in which the term with diffusion constant (*D*) has to be included:(1)$$\frac{\partial n}{\partial t}=D\frac{\partial^{2}n}{\partial z^{2}}-{(k}_{1}+k_{IET}+k_{IHT})\;n-k_{2}n^{2}-k_{3}n^{3}.$$
The above-resultant equation can only be solved numerically to properly describe the interfacial electron- and hole-transfer dynamics and extract the charge transfer parameters. This is probably the reason why it is still rarely used in the analysis of time-resolved measurements of PSCs on the time scale from ps to ns. 

Several analytical functions are frequently used to model the decay of the charge population observed in time-resolved measurements of PSCs, and they are critically outlined below. The first one is the multi-exponential function [17,18]:(2)$$∆A\left(t\right)=A_{0}+A_{1}e^{-\frac{t}{\tau_{1}}}+A_{2}e^{-\frac{t}{\tau_{2}}}+\cdots +A_{n}e^{-\frac{t}{\tau_{n}}},$$
with time constants $\tau_{i}$ related to pre-exponential factors $A_{i}$, and $A_{0}$ being an optional constant offset component. In the simplest way, the multi-exponential functions are fitted at a single wavelength (e.g., at a maximum of emission or a minimum of bleach signal in transient absorption) [17,19,20]. In broadband measurements, much more information can be gathered via global fitting in the broad spectral range [18,21,22,23,24,25,26,27]. In the global fitting using the multi-exponential function, the pre-exponential factors $A_{i}$ in Equation (2) become wavelength-dependent and are called pre-exponential factor spectra or decay-associated spectra (DAS). Increasing the number of exponential components can result in a very good quality of fit. However, the particular time components usually do not have a certain physical meaning because almost no processes in perovskites can be described by exponential function decay. In particular, higher-order recombination and/or diffusion cannot be represented by the exponential functions. 

The differences in the pre-exponential factor spectra sometimes enable us to distinguish different processes in the charge population evolution, and the corresponding time constants can be then treated as the average time scale of these processes. For example, despite poor fit quality, we used two-exponential approximation in the global analysis of broadband transient absorption data to separate charge cooling in PSCs (time-constant typically <1 ps) from the average decay of the bleach band (times from hundreds of ps to single ns) [18,21,22]. The addition of more exponential components, especially for the samples excited at relatively high pump fluence, revealed bleach decay with the bands blue-shifted for shorter time constants, confirming that the band-filling mechanism in perovskites affects the more short-wavelength transitions at shorter times [21,28]. This is one of the justifications why broadband bleach integration is necessary for the proper representation of the photoinduced charge population (see below).

Another analytical approach concentrated on the important role of second-order recombination in charge population decay. When the diffusion term and third-order recombination are neglected (D=0, $k_{3}=0$), then Equation (1) can be solved analytically, leading to the so-called mixed first and second-order decay function:(3)$$∆A\left(t\right)=\frac{A\;k_{1}}{{k_{1}e}^{k_{1}t}-k_{2}n_{0}\;\left(e^{k_{1}t}-1\right)},$$
where $n_{0}$ is the number of excited carriers at t=0 and A is the proportional term. The function above is mostly employed for fitting time-resolved emission data [29,30,31]. However, we also tried it for transient absorption results. The fit quality was sufficient in certain cases but mostly at low excitation fluence [21]. The typical verification of this model is to check whether the first and second-order rate constants ($k_{1}$ and $k_{2}$) do not change in the fit at different excitation fluences.

Finally, the stretched-exponential function is sometimes used for fitting the charge decays in PSCs:(4)$$∆A\left(t\right)=A_{0}e^{-{\left(\frac{t}{\tau }\right)}^{\beta }},$$
with the stretched parameter 0<β≤1 (at lower values, the function is “spread” over more time scales). This function can describe signal decay as a distribution of sources decaying at different rates, e.g., due to diffusion. The physical meaning can have the averaged decay time ($\tau_{AVG}$), which can be obtained as
(5)$$\tau_{AVG}=\frac{\tau }{\beta }\Gamma \left(\frac{1}{\beta }\right),$$
where Γ is the gamma function. Time-resolved data of PSCs are successfully fitted with the stretched-exponential function in a limited number of cases [21,22], mostly at a low pump fluence. Interestingly, we recently found an improved fit quality when the mixed first- and second-order function Equation (3) is additionally stretched (β=0.3) [22]; however, its physical meaning is hard to explain.

It should be stressed that although all the above analytical models can offer some clues regarding the time scales of the charge transfer processes in perovskites, they cannot provide the precise values of the rate constants because the charge diffusion is not included in any of them. In particular, the interfacial charge injection rate constants that are extracted from such models are significantly smaller than the values of $k_{IET}\;$ and $k_{IHT}$ fitted in the numerical simulation of Equation (1). Therefore, in this work, we show how Equation (1) can be used to study the ultrafast and fast charge dynamics in complete PSCs (with ETM, HTM, and both electrodes) and discuss its interesting implications.

A versatile way to study fast charge population decay in PSCs is to analyze the bleach dynamics observed in transient absorption (Δ*A*, measured in the transmission mode) upon pulse excitation. Such transient absorption studies permit the easy collection of the decays for several pump pulse fluences, selective excitation from both ETM and HTM sides, and, as we recently showed, even the measurements through metal electrodes under an applied bias voltage [32]. In some reports about PSCs, it is assumed that single-electron injection to ETM or separate hole transfer to HTM does not contribute to bleach recovery, probably in analogy to most of the organic (molecular) systems in which the charges are localized and the simultaneous extraction of both electron and hole (from a photoinduced electron–hole pair) is necessary to observe ground state recovery (bleach decay). However, in perovskites, like in other inorganic semiconductors, the electrons and holes exist as free charges and are delocalized. Therefore, the changes in their population due to transfers to contact materials can be independently observed via bleach band decay (see Appendix A). This feature has been frequently confirmed in PSCs transient absorption studies in which the bleach decay accelerated after the addition of contact material (single ETM or HTM) with respect to pristine perovskite or by recording bleach kinetics similar to fluorescence decay [6,13,18,23,24,33]. 

The inclusion of the diffusion term in Equation (1) has been sometimes used in the studies of isolated perovskite materials [12,15,23,34] but very rarely in the systems with the ETMs and/or HTMs. In one of the pioneering works [16], MAPbI_3_ perovskite of different thicknesses with the contacts of either PCBM or spiro-OMeTAD was studied, and the intrinsic interfacial electron and hole transfers were observed to be much faster than the diffusion process. Moreover, it has also been shown that electrons and holes contribute with different weights to the bleach signal [16]: (6)$$∆A\left(t\right)\propto \chi n_{e}\left(t\right)+\left(1-\chi \right)n_{h}\left(t\right),$$
where *n*_e_*, n*_h_*—*density of electrons and holes and *χ*—the weight of electron contribution to the bleach feature (fitted value: *χ* = 0.7). We based our methodology on the above approach, but we extended it to the complete solar cells (with both ETM and HTM), included Auger recombination, and proposed an alternative way to correctly determine charge population based on band integral (*BI*) instead of single Δ*A* kinetics at bleach minimum [21].

## 2. Results and Discussion

Our methodology of charge-transfer modeling is described below. Equation (1) treats electrons (*n*_e_) and holes (*n*_h_) separately, as they exist as free charges in perovskites, offering the following set of equations used for bleach decay modeling [16,35]:(7)$$\frac{\partial n_{e}}{\partial t}=D\frac{\partial^{2}n_{e}}{\partial z^{2}}-k_{1}n_{e}-k_{2}n_{e}n_{h}-k_{3}{n_{e}n_{e}n}_{h}-\delta_{e}\left(z\right)k_{IET}n_{e},$$
(8)$$\frac{\partial n_{h}}{\partial t}=D\frac{\partial^{2}n_{h}}{\partial z^{2}}-kk_{1}n_{h}-k_{2}n_{e}n_{h}-k_{3}{n_{e}n_{h}n}_{h}-\delta_{h}\left(z\right)k_{IHT}n_{h}.$$
Both charge distributions *n*_e_ and *n*_h_ are the functions of time (*t*) and distance (*z*) in perovskite. Intrinsic electron and hole transfers with rate constants *k*_IET_ and *k*_IHT_ occur close to the interface (in our model, at a 1 nm distance from ETM or HTM); thus, delta functions are zero except for $\delta_{e}=1$ for 0 < *z* < 1 nm and $\delta_{h}=1$ for *L*—1 nm < *z* < *L*, where *L* is the perovskite thickness (Figure 1B). Calculation of proper charge population requires broad spectral integration over the bleach band to take into account the bleach broadening effect due to band filling mechanism [21,22,32,36]:(9)$$BI\left(t,\Delta \lambda =\lambda_{2}-\lambda_{1}\right)=\int_{\lambda_{1}}^{\lambda_{2}}\Delta A\left(t,\lambda \right)\frac{d\lambda }{\lambda }\;.$$
The above so-called band integral (*BI*) amplitude taken from the transient absorption measurements is thus proportional to the weighted contribution of electron and hole populations:(10)$$BI\left(t\right)=S\left(\chi \int_{0}^{L}n_{e}\left(z,t\right)dz+\left(1-\chi \right)\int_{0}^{L}n_{h}\left(z,t\right)dz\right),$$
where *S* is the scaling factor. The initial distributions of the charges are as follows (Figure 1B):(11)$$n_{e}\left(z,0\right)=n_{h}\left(z,0\right)=n_{0}\exp\left(-\alpha z\right),$$
where *n*_0_ = α *J*, α is the absorption coefficient (in cm^−1^), *J* is pump fluence (in photons/cm^2^).

The simulations of the system were performed in COMSOL Multiphysics software v. 5.4 (see Section 3). The expression in the brackets of Equation (10) is called weighted surface charge distribution (WSCD) and presented as a function of time in the units of cm^−2^ in most of the figures below.

First, we validated the proposed model by comparing it with the experimental results. Figure 2 shows the representative examples for our PSCs made using the benchmark triple-cation, mixed halide lead perovskite (FA_0.76_MA_0.19_Cs_0.05_Pb(I_0.81_Br_0.19_)_3_) with standard mesoporous TiO_2_ as ETM and spiro-OMeTAD as HTM layers, showing the effects of different pump pulse fluences, different applied voltages, different excitation sides (ETM and HTM) as well as different excitation wavelengths (thus different absorption coefficients of the perovskite layer). The samples were obtained from our previous reports [28,32], and the new ones were prepared in an analogous way. The triple-cation mixed halide perovskite composition is one of the most stable perovskite formulas, especially for humidity and structural impurities, with very high PCE [37]. The *BI* over the spectral range from $\lambda_{1}=$ 680 nm to $\lambda_{2}=$ 800 nm was taken to cover the entire bleach range (including that at the initial time delays; see Appendix A). In each part of Figure 2A–D, the fits are global in the sense that such common values of parameters are searched for which the different curves are simultaneously well reproduced. As can be seen, the agreement between the experimental data and the numerical calculations is good, e.g., the Pearson correlation coefficients for the two curves in Figure 2A are 0.997 and 0.996. The extracted parameters (diffusion coefficient and rate constants) are in the following range for the different samples in each figure (Table 1): *D* = (1 ÷ 3) × 10^−2^ cm^2^/s, *k*_1_ = 3 × 10^5^ s^−1^, *k*_2_ = (0.8 ÷ 3) × 10^−10^ cm^3^/s, *k*_3_ = (0.2 ÷ 7) × 10^−28^ cm^6^/s, *k*_IET_ = (1 ÷ 3) × 10^11^ s^−1^, *k*_IHT_ = 1 × 10^11^ s^−1^; χ was fixed at a value of 0.7 based on reference [16]. These values are in general agreement with other reports for PSCs, although the time-resolved reports of the studied triple-cation perovskites are quite rare [22,32,38], so the exact comparison is difficult. 

The high rates of intrinsic charge injections to contact material (*k*_IET_ and *k*_IHT_, injection times in single ps range) should be highlighted, confirming that the apparent slow bleach decay over single ns is due to diffusion contribution [16]. Appendix A presents the simulation of the distribution of the charges across the perovskite thickness at different time delays and different pump fluences, showing the interplay between charge diffusion, charge recombination, and charge injection. The necessity of the inclusion of the Auger recombination (rate constant *k*_3_) to properly describe the experimental data should be stressed. The contribution of the third-order recombination was neglected in many of the previous time-resolved studies on PCSs at similar pump fluence. We think that it might be due to the fact that the bleach kinetics analysis has been earlier limited only to its minimum. In this way, the initial fast decay that repopulates mainly the high energetic (short-wavelength) part of the bleach spectrum is missing, and the kinetics are apparently slower. Its inclusion in the proper spectral integration (*BI*) makes the charge population decay faster [21]. Next, it can be noticed that the residual signal is smaller for the excitation from HTM than the ETM side (Figure 2C), which we previously interpreted as a possible faster hole injection to HTM than electron injection to ETM [22]. However, in the current model, the fitted injection rate constants are found to be the same (*k*_IET_ = *k*_IHT_), and the difference occurs due to the asymmetry in the contribution of electrons and holes to bleach decay (χ > 0.5). The modeled first-order recombination (*k*_1_) is kept at low values (µs time range), so it does not contribute to the charge population decay, while for the pump fluence of several tens of µJ/cm^2^, the decay is dominated by *k*_2_ and *k*_3_. Obviously, it leads to the acceleration of the charge population decay upon increasing the pump fluence (Figure 2C, or comparison between Figure 2A,B). Interestingly, for the samples with better photocurrent (Figure 2C), the recombination rate constants *k*_2_ and *k*_3_ are smaller than those for the samples with worse photocurrent (Figure 2B). Finally, tuning the excitation wavelength towards the red makes the apparent kinetics slower (with all rate constants and diffusion coefficient being the same, Figure 2D). This is because the absorption coefficient α decreases, and the initial charge distribution in Equation (11) becomes more uniformly distributed along the perovskite depth *z*, so the charge recombination and interfacial injection are less probable, and the lifetime of the charges increases.

We are aware that the model could be further extended, taking into account self-absorption (of emitted photons), additional interfacial charge recombination processes, different diffusion constants for electrons and holes, and/or distance-dependent *k*_1_ and *k*_2_ rate constants (different close to both interfaces) [15,22,35,38]. Moreover, drift and capacitive charging effects have been shown to be essential to model the charge decay on longer time scales, from tens of nanoseconds to microseconds (using combined drift–diffusion and Poisson equations) [39,40]. The approximation of an exponential decay of the absorption profile given by Equation (11) might not be sufficient if the perovskite thickness is smaller and/or the excitation wavelength is longer (smaller α values) than those used in our studies. Then, all excitation light will not be absorbed within the perovskite layer and additional effects due to light reflection at opposite interface and light interference should be taken into account, e.g., using transfer matrix methods [41,42]. The mixed halide perovskite that we used in our studies might also suffer from photoinduced ion migration that can be a source of additional cation-mediated recombination and affect the charge transfer kinetics and hysteresis of the cells [43,44]. We have recently observed the photoinduced changes in transient absorption spectra and kinetics for our triple cation mixed halide samples when the transient absorption experiment lasted more than several minutes [32]. Therefore, the experimental results modeled in the current studies were obtained from short-time measurements of fresh samples when the ion-migration effects could be neglected. We also do not consider the possible electron or hole injections during sub-ps charge cooling and accompanying exciton dissociation processes. However, for the examples tested by us so far, the agreement with the experimental data is sufficient on a time scale from single ps to single ns, while adding more parameters makes the fitting more complicated and less unequivocal. 

Our current model enables us to reveal several interesting and apparent relationships between the parameters, which we discuss below based on the performed additional simulations. Figure 3, Appendix A show the results of the simulated parameter variations. We performed it for a 475 nm excitation (α = 190,000 cm^−1^) at two pump pulse fluences: high (30 µJ/cm^2^) and low (2.3 µJ/cm^2^), corresponding to *n*_0_ = 1.3 × 10^19^ cm^−3^ and *n*_0_ = 0.1 × 10^19^ cm^−3^, respectively, which represent the experimental conditions from Figure 2. The initial values of the diffusion and rate constant parameters are also similar to those determined from the above fits to the experimental *BI* decays.

First, it should be noticed that at a short-wavelength excitation (<500 nm) and the perovskite thickness *L* > 500 nm, there is no influence of the further *L* increase on the charge population kinetics (up to 3 ns, which is the temporal window of our setup; Figure 3A) and it is independent from the charge injection to the opposite side contact material. For example, for the excitation from the ETM side, the kinetics become faster with increasing the electron injection rate constant (for *k*_IET_ > 1 × 10^9^ s^−1^, Figure 3B and Appendix A) while they do not change at all for any variation of the hole injection rate constant (*k*_IHT_, Appendix A). The situation is fully symmetric on the HTM side (no influence of *k*_IET_); only the residual signal is different if χ is different from 0.5. It confirms that one can probe selectively different contacts (ETM and HTM) by the excitation from different sides of the perovskite, which is a great advantage of femtosecond transient absorption setups over many other techniques for PSC characterization. 

Next, for high pump pulse fluence (>5 µJ/cm^2^), the dominant contribution to the charge population decay comes from the second-order (*k*_2_) and even third-order (*k*_3_) recombination (Figure 3C and Appendix A) and the effect of *k*_IET_ (or *k*_IHT_) variation is small (Figure 3B). For 30 µJ/cm^2^, the residual signal at 3 ns is greatly dependent on *k*_2_ values, while *k*_3_ modifies the kinetic profile at short times (tens of ps)—see Appendix A. On the contrary, at lower fluence, the kinetics significantly depend on the interfacial charge injection rate constants (Figure 3B), the contribution of *k*_2_ is decreased (Appendix A), while the effect of *k*_3_ is marginal (Appendix A). This suggests that the possibility of extraction of the precise values of charge injection rate constants (*k*_IET_ and *k*_IHT_) is limited under high pump fluence. Thus, longer-lasting transient absorption experiments at lower pump fluencies are necessary. However, a decrease of the pump fluence much below 1 µJ/cm^2^ does not have much effect on the kinetics, since higher-order recombination is already significantly suppressed below 5 µJ/cm^2^ (Figure 3C). It can also be noted that first-order recombination starts to offer some contribution to the simulated decays for values *k*_1_ = 10^8^ s^−1^ or higher (Appendix A).

Furthermore, at certain pump fluence, the increase in the residual value of the *BI* kinetics (at 3 ns) can be realized via the variation of the weight of electron contribution (χ) or the variation of interfacial injection rate constant (e.g., *k*_IET_ for the excitation from the ETM side). Figure 3B, Appendix A show examples of the population decay for several values of these two parameters at low pump fluence. Therefore, as we mentioned before, a simple comparison of the residual values for the excitation at ETM and HTM sides cannot unambiguously lead to the comparison of electron and hole injection rates (Figure 2C). For example, Figure 3D shows that for the excitation from the ETM side, the same increase in residual value is realized when *k*_IET_ drops from 1 × 10^11^ s^−1^ to 1 × 10^10^ s^−1^ or when χ decreases from 0.7 to 0.3. However, the shape of kinetics is different when the residual value increases due to the change in *k*_IET_ (faster initial decay) or due to the change in χ (slower initial decay). Thus, with a sufficient signal-to-noise ratio of the experimental data, the fit of our model should distinguish these two cases. 

Nevertheless, we propose that the most reliable method for *k*_IET_ and *k*_IHT_ determination is transient absorption measurements without and with applied forward bias voltage (at the value close to the open circuit voltage of the cell, V_OC_, so there is no current flow in the cell) and exciting the cells from ETM and HTM sides to determine separately *k*_IET_ and *k*_IHT_, respectively. In this case, one can be sure about the same perovskite morphology and properties both with charge injections (*k*_IET_ > 0 or *k*_IHT_ > 0) and when these processes are blocked by bias forward voltage of about *V*_OC_ (then *k*_IET_ = *k*_IHT_ = 0). The lack of perfect fit in Figure 2C may suggest that the material parameters (*D*, *k*_1_, *k*_2_, *k*_3_, χ) might be slightly different at ETM and HTM sides, and it is better to make the global fit for the excitation from only one side (preferably at several pump fluences). We recently proposed a way to measure the transient absorption of PSCs through the metal electrodes and, thus, under applied voltage [32]. In that previous work, we showed the effect of forward *V*_OC_ bias at relatively high fluence, so the changes in the kinetics due to the influence of *k*_IET_ were small (e.g., residual signal relative change of about 5% between 0 V and V_OC_ bias, Figure 2A). For the current work, we made new PSCs and repeated the analogous measurements under a lower pump fluence at which the influence of *k*_IET_ is more pronounced (residual signal change of about 18%; Figure 2B). In both cases (Figure 2A,B), our assumptions are supported by good model fits. Furthermore, we also tested the influence of different forward biases and found that the value of the applied voltage around *V*_OC_ is not critical—in our case, the *BI* kinetics were similar between −900 and −1200 mV (Appendix A).

Finally, the interesting effects of diffusion coefficient variation can be pointed out (Appendix A). When *D* increases from 0 to 0.02 cm^2^/s, the charge population decay becomes faster, because more charges are injected. However, further increase in *D* (up to the simulated values of 2 cm^2^/s) results in slower decay in the 3 ns time window, because electrons and holes are spreading faster over the perovskite volume and the probability of second- and third-order recombination decreases. 

The above model has discrete boundary conditions (delta functions) that open the electron or hole injection channels at 1 nm distance from the contacts with ETM (*z* = 0) or HTM (*z* = *L*), respectively. The charge injections rate constants (*k*_IET_, *k*_IHT_) have, then, the experimental units of s^−1^, identical to *k*_1_. In the model without diffusion, it is a reasonable way to introduce the charge injection on the boundary. However, arbitrarily choosing of the injection range is required, which, when changing its value, leads to the change in the injection rate constant. The use of diffusion enables a natural way for the implementation of interfacial charge transfer using the boundary conditions. This approach does not allow any arbitrary assumptions. Therefore, the boundary conditions should take the form of the following equations [34,35]: (12)$${\left.D\frac{\partial n_{e}}{\partial z}\right|}_{z=0}=S_{IET}n_{e};\;\;{\left.D\frac{\partial n_{h}}{\partial z}\right|}_{z=L}=-S_{IHT}n_{h},$$
where parameters *S*_IET_ and *S*_IHT_ have the units of cm/s and can be called the speed of electron and hole injection, respectively. The relation between the parameters can be calculated from formula $S_{IET/IHT}=k_{IET/IHT}\times width\;of\;\delta_{e/h}(z)\;area=k_{IET/IHT}\times 10^{-7}cm$. We tested the above correction and we found that both approaches (discrete and continuous) are in agreement (Appendix A). The implementation of the boundary conditions is presented in SI.

## 3. Materials and Methods

### 3.1. Sample Preparation

The preparation protocol followed that published by Saliba et al. [45]. Plates of the size 2.4 × 2.4 cm of FTO (fluorine-doped tin oxide) substrates (FTO glass, ≈13 Ω/sq, Sigma-Aldrich, Merck KGaA, Darmstadt, Germany) were cut, and part of the FTO conductive layer was etched by Zn powder and HCl to prevent charge recombination between gold electrodes and FTO. The etched substrates were brushed with detergent (Hellmanex, Hellma, Germany) and then bathed in detergent, distilled water, and isopropanol in ultrasounds, each step of 15 min. Finally, a UV ozone cleaner was used for 15 min. A compact TiO_2_ layer was deposited by spray pyrolysis at 450 °C using titanium diisopropoxide in ethanol (1:14 *v*/*v* in EtOH) as a precursor solution. A mesoporous titania layer was obtained by deposition of titania paste (30NR-D, GreatCell Solar, Queanbeyan East, Australia, diluted 1:6 *w*/*w* in EtOH) by spin coating (10 s, 2000 rpm) and annealing the substrate at 500 °C for 30 min. FAPbI_3_/MAPbBr_3_/CsI precursor solutions were prepared in glovebox conditions (in nitrogen flux) and mixed to obtain a 1.5 M precursor solution of the FA_0.76_MA_0.19_Cs_0.05_Pb(I_0.81_Br_0.19_)_3_ perovskite. The solution was spin-coated on a mesoporous layer in a dry box with an additional nitrogen flow for 10 s at 2000 rpm and then for 20 s at 4000 rpm; 10 s before the end of this process, chlorobenzene as an anti-solvent was used. A solution of 2,2′,7,7′-tetrakis-(N,N-di-4methoxyphenylamino)-9,9′-spirobifluorene (Spiro-OMeTAD, Sigma-Aldrich) (72.3 mg/mL in chlorobenzene) with additives (17.5 µL/mL of a 520 mg/mL LiTFSI solution in acetonitrile and 28.8 µL/mL *4-tert-*buthylpyridine) was also spin-coated (4000 rpm, 30 s). Finally, 70 nm gold electrodes were sputtered on the top.

### 3.2. Transient Absorption

Femtosecond transient absorption spectroscopy studies were performed on a 1 kHz repetition rate and a 0.4 ps response function setup delivered by Ultrafast Systems (Spectra-Physics laser system, Santa Clara, CA, USA, and Helios spectrometer, Sarasota, FL, USA) [32]. Measurements were performed in the Vis-NIR (500–850 nm) region, with a 475 nm or 495 nm excitation wavelength, in the temporal window of up to 3 ns. Additional bias during transient absorption measurement was applied to the electrodes by a compact potentiostat (model PGSTAT204, Metrohm Autolab, Utrecht, The Netherlands).

### 3.3. Numerical Simulations

Simulations were performed in COMSOL Multiphysics software (version 5.4). In addition, 1D geometry was used. The perovskite layer was added as an Interval from 0 to *L*. In the Physics interface, Coefficient Form PDE from the Mathematics module was used to implement Equations (7) and (8), solving two dependent variables *n*_e_ and *n*_h_. The implementation consisted of the following parameters defined in Coefficient Form PDE:$$c=\left(\begin{array}{cc} D & 0\\ 0 & D \end{array}\right),$$
$$a=\left(\begin{array}{cc} k_{1}+k_{2}n_{h}+k_{3}n_{e}n_{h}+k_{IET}\delta_{e}(z) & 0\\ 0 & k_{1}+k_{2}n_{e}+k_{3}n_{e}n_{h}+k_{IHT}\delta_{h}(z) \end{array}\right),$$
$$d_{a}=\left(\begin{array}{cc} 1 & 0\\ 0 & 1 \end{array}\right).$$

The remaining parameters were equal to zero. $\delta_{e}$ and $\delta_{h}$ were implemented using the Step function. The initial distribution of charges was implemented in Initial Values as $n_{e}=n_{0}\exp\left(-\alpha z\right)\;\left[n_{0}\exp\left(-\alpha \left(L-z\right)\right)\right]$ and $n_{h}=n_{0}\exp\left(-\alpha z\right)\;\left[n_{0}\exp\left(-\alpha \left(L-z\right)\right)\right]$ for the excitation from the ETM (HTM) side. The mesh with the element size of 0.05 nm was used. Simulations were performed using a Time Dependent study with predefined settings. The simulation data were collected from 1 ps to 3 ns using the logarithmic step with 50 steps per decade.

The boundary conditions defined in Equation (12) were implemented in Coefficient Form PDE using the Flux/Source function. Here, parameter $S_{IET}$ was implemented as
$$q=\left(\begin{array}{cc} S_{IET} & 0\\ 0 & 0 \end{array}\right)$$
at the boundary placed at z=0 and $S_{IHT}$ as
$$q=\left(\begin{array}{cc} 0 & 0\\ 0 & S_{IHT} \end{array}\right)$$
at the boundary placed at z=L, while term p was equal to zero. The constants used in the study ($k_{1},k_{2},k_{3},k_{IET},k_{IHT},D,n_{0},\alpha ,L,\chi ,S_{IET},S_{IHT}$) were defined in Global Definitions as Parameters.

## 4. Conclusions

In conclusion, we propose a simple mathematical model and methodology for the extraction of correct values of the parameters that govern charge transfers in complete PSCs on ultrafast and fast time scales (ps–ns). We refine the existing approaches and show the necessity of including the charge diffusion and both second- and third-order charge recombination processes in the interpretation of electron and hole population kinetics. Broadband analysis of transient absorption data is also required. For sufficiently short-wavelength excitation and the perovskite layer thickness typical for complete cells (>500 nm), the charge injection at only one (incident) interface is important (perovskite/ETM or perovskite/HTM). The contribution of electron or hole injection can be most easily observed at low excitation fluences (<5 µJ/cm^2^) by comparing the data with and without the applied bias voltage. We believe that our work will pave the way for more properly analyzed time-resolved studies of the interfaces between perovskites and selective contact materials leading to a better understanding of the factors that influence the efficiency of PSCs.

## Figures and Tables

**Figure 1 materials-16-07110-f001:**
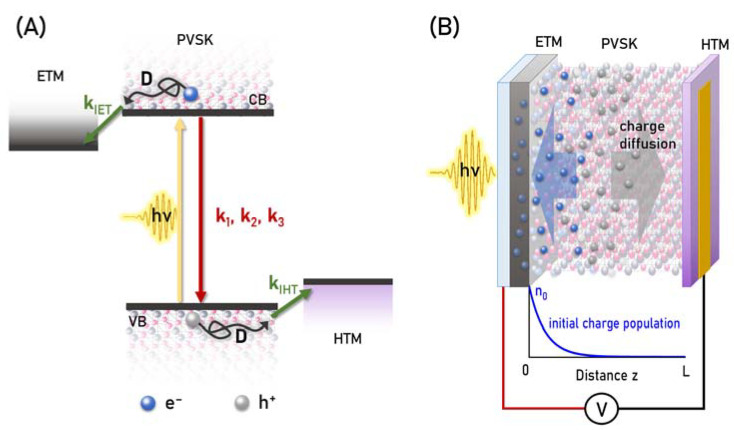
(**A**) Scheme of the energy levels and charge-transfer processes in PSCs. (**B**) Scheme of the configuration of PSC for the measurements under applied voltage for pulse excitation from the ETM side; bottom: initial charge distribution along perovskite thickness.

**Figure 2 materials-16-07110-f002:**
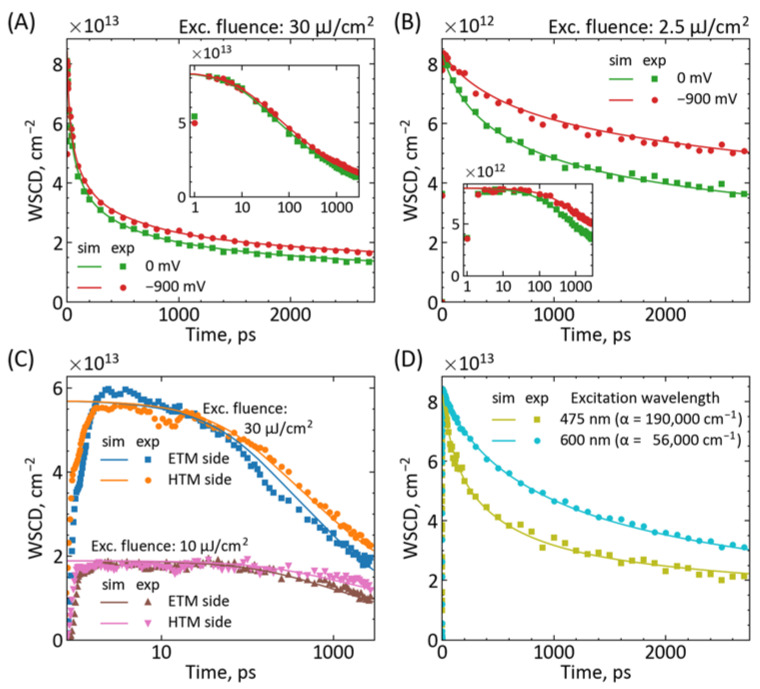
Comparison between the experimental results (exp, points, from *BI*) and numerical modeling (sim, lines) based on Equation (10). The solar cell and modeling fitted parameters are given in Table 1. (**A**) Our recently published experimental results for different bias potentials (0 V and −900 mV) [32]. In the latter case, the current flow is blocked, so charge injection is not possible (*k*_IET_ = 0 s^−1^ for −900 mV). (**B**) The new results at a lower pump fluence for 0 V and −900 mV bias potentials (*k*_IET_ = 0 s^−1^ for −900 mV). (**C**) The experimental results for our highest-efficiency PSC device [28] measured at two pump fluences and from both ETM and HTM sides. (**D**) The effect of pump wavelength, the excitation from the HTM side (through the gold electrodes). We note the logarithmic scale on the horizontal (time) axis in (**C**) and the insets in (**A**,**B**). The effect of possible photoinduced changes in the samples [32] was monitored and minimized.

**Figure 3 materials-16-07110-f003:**
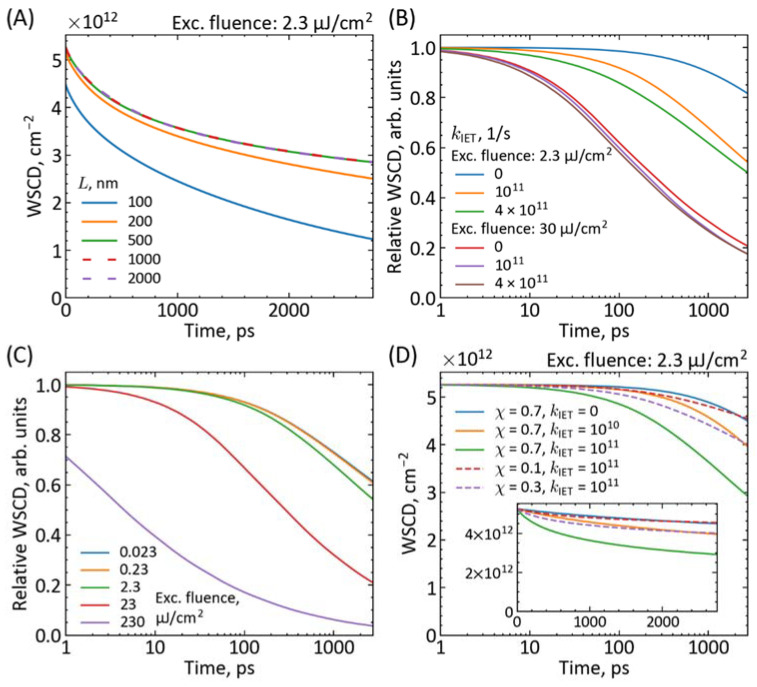
Simulations of charge population decay observed in transient absorption of PSCs (calculated from Equation (10)) for different parameter variations: (**A**) perovskite thickness, (**B**) electron injection rate constant, (**C**) pump fluence, (**D**) electron injection rate constant and electron contribution to the bleach. If not otherwise indicated in the graph insets, the values of the parameters in the simulations are as follows: *L* = 500 nm, χ = 0.7, *D* = 2 × 10^−2^ cm^2^/s, *k*_1_ = 1 × 10^5^ s^−1^, *k*_2_ = 2 × 10^−10^ cm^3^/s, *k*_3_ = 2 × 10^−28^ cm^6^/s [0 for (*D*)], *k*_IET_ = *k*_IHT_ = 1 × 10^11^ s^−1^, α = 190 000 cm^−1^ (475 nm excitation from the ETM side). Pump fluence of 2.3 µJ/cm^2^ corresponds to *n*_0_ = 0.1 × 10^19^ cm^−3^.

**Table 1 materials-16-07110-t001:** Solar cell and charge transfer parameters of the modeling fitted to the experimental presented in Figure 2.

Parameter	Figure 2A	Figure 2B	Figure 2C	Figure 2D
Pump (side, fluence)	ETM side, 30 μJ/cm^2^	ETM side, 2.5 μJ/cm^2^	Both sides, 10 and 30 μJ/cm^2^	HTM side, 30 μJ/cm^2^
*L*, nm	500	500	600	500
*D,* cm^2^ s^−1^	3 × 10^−2^	3 × 10^−2^	1 × 10^−2^	2 × 10^−2^
*k*_1_, s^−1^	3 × 10^5^	3 × 10^5^	3 × 10^5^	3 × 10^5^
*k*_2_, cm^3^ s^−1^	0.9 × 10^−10^	3 × 10^−10^	0.8 × 10^−10^	3 × 10^−10^
*k*_3_, cm^6^ s^−1^	2.5 × 10^−28^	7 × 10^−28^	0.4 × 10^−28^	0.2 × 10^−28^
*k*_IET_, s^−1^	3 × 10^11^ or 0	1 × 10^11^ or 0	1 × 10^11^	1 × 10^11^
*k*_IHT_, s^−1^	-	-	1 × 10^11^	-
*J*_sc_, mA cm^−2^	20	15	24	20

## Data Availability

The data presented in this study are available on request from the corresponding author.

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
