# Peer review of "Modeling of Charge Injection, Recombination, and Diffusion in Complete Perovskite Solar Cells on Short Time Scales"

_materials, 2023, doi:10.3390/ma16227110_

Round 1

Reviewer 1 Report

Comments and Suggestions for Authors

In this manuscript, the authors proposed a model of charge population decay upon ultrafast optical pulse excitation in complete,  working perovskite solar cells. The equation including charge injections to contact materials, charge diffusion, and charge recombination via first, second, and third-order processes is solved using numerical simulations. I commend the authors for their work and I recommend the manuscript published with the following minor revisions. Please check:

1. The relative references about equation 2 is suggested.

2. Why do the authors choose the mixed halide lead perovskite of FA0.76MA0.19Cs0.05Pb(I0.81Br0.19)3? 

3. The relative references about equation 12 is also suggested.

4. The full name of FTO should be given when it first appears.

Reviewer 2 Report

Comments and Suggestions for Authors

The authors implement a numerical model based on a rate equation to describe charge injection, recombination and diffusion at fast and ultrafast time scales in perovskite solar cells (PSCs). The calculations are performed using COMSOL and compared with experimental data based on transient absorption spectroscopy performed on PSCs with a standard configuration. From fitting the experimental data the parameters governing the charge dynamics are extracted, such as the diffusion coefficient and rate constants.

The motivation for this study is to introduce an easy-to-use framework based on COMSOL software, which includes higher-order processes like radiative and Auger recombination. In this context the authors provide a rather detailed description of semi-analytical approaches usually employed. The current approach may provide a bridge towards a more accurate interpretation of charge kinetics in PSCs, especially for supporting experimental procedures.

I would recommend the publication of this paper, with two amendments:
- comment on the limitations of the current approach;
- detail the input and procedures used for COMSOL simulations (in the Supporting Information), so that they are easily accessible;

Reviewer 3 Report

Comments and Suggestions for Authors

The manuscript „Modeling of Charge Injection, Recombination, and Diffusion in Complete Perovskite Solar Cells” by Szulc et al. presents a transient drift-diffusion model describing the decay of charge population after optical excitation. The implemented model is applied to the transient absorption results of state-of-the-art triple cation perovskites, capturing the charge transfer dynamics accurately.

The given study is certainly interesting, as it seems that a simple model can capture the transient properties nicely. As a large part of the motivation is focused on full solar cell devices – which the model does not capture – some open questions remain that must be addressed in a revision.

1.      Charge transport layers are modeled as ideal sinks of charge carriers, assuming that - upon transfer to the transport layers – charges are extracted ideally. The wording “charge injection rate constant” is misleading, as they suggests that charges are injected into the perovskite layer. Considering equations 7 and 8, it is clear that charges are extracted only and not injected into the perovskite layer.
This should be rephrased and clearly stated. The potential impact of neglecting charge accumulation at the HTL  or ETL side should be further discussed.

2.      Previous studies considered the migration of halide interstitials/vacancies within DD models and showed a relevant impact of their migration on the recombination properties [e.g. 10.1039/c8ee01576g; 10.1016/j.solmat.2020.110912]. As MHPs are prone to ion migration and accumulation, the role of screening from ions to the recombination of electrons and holes should be discussed in detail.

3.      For thin layers, the approximation of an exponential decay of the absorption profile might be a strong approximation and change the outcome on the relevance of the transport layers. The use of an optical transfer matrix method that accounts for interference and scattering effects has been used in recent years, see especially the work of Burkhard et al. [10.1002/adma.201000883]. I would appreciate some comparison with the TMM method.

4.      The model works only at open-circuit conditions, at which a flat potential may be assumed. This condition is useful to reproduce experimental results on absorption and emission properties of thin films but does not model a full device. Thus, the title “Modeling … complete perovskite solar cells” is not fulfilled in this paper and must be changed, and consequently also the manuscript text adapted. Alternatively and even better, realistic operating conditions could be implemented. 

Round 2

Reviewer 3 Report

Comments and Suggestions for Authors

The authors addressed all comments in detail. The focus on the short timescales is convincing to justify the validity of the results on full perovskite solar cell devices. The paper can be published in its current form.